# Design and Implementation of Two Immersive Audio and Video Communication Systems Based on Virtual Reality

Hanqi Zhang [1], Jing Wang [1,*], Zhuoran Li [2,*] and Jingxin Li [3]

1   School of Information and Electronics, Beijing Institute of Technology, Beijing 100811, China
2   Aerospace Information Research Institute, Chinese Academy of Sciences, Beijing 100045, China
3   China Electronics Standardization Institute, Beijing 101102, China
*   Correspondence: wangjing@bit.edu.cn (J.W.); lizhuoran@aircas.ac.cn (Z.L.)

**Abstract:** Due to the impact of the COVID-19 pandemic in recent years, remote communication has become increasingly common, which has also spawned many online solutions. Compared with an in-person scenario, the feeling of immersion and participation is lacking in these solutions, and the effect is thus not ideal. In this study, we focus on two typical virtual reality (VR) application scenarios with immersive audio and video experience: VR conferencing and panoramic live broadcast. We begin by introducing the core principles of traditional video conferencing, followed by the existing research results of VR conferencing along with the similarities, differences, pros, and cons of each solution. Then, we outline our view about what elements a virtual conferencing room should have. After that, a simple implementation scheme for VR conferencing is provided. Regarding panoramic video, we introduce the steps to produce and transmit a panoramic live broadcast and analyze several current mainstream encoding optimization schemes. By comparing traditional video streams, the various development bottlenecks of panoramic live broadcast are identified and summarized. A simple implementation of a panoramic live broadcast is presented in this paper. To conclude, the main points are illustrated along with the possible future directions of the two systems. The simple implementation of two immersive systems provides a research and application reference for VR audio and video transmission, which can guide subsequent relevant research studies.

**Keywords:** virtual reality; immersive communication; VR conference room; panoramic live broadcast; spatial audio

## 1. Introduction

Virtual reality (VR) technology has gone through numerous periods since it was first proposed [1,2]. Currently, applications of virtual reality can be seen in many fields, such as entertainment, education, medical care [3], real estate, etc. [4–6]. All of these benefit from its ability to provide a sense of immersion. The promotion of the concept "Metaverse" has ushered in another peak. The future application scenarios of VR include an end-to-end immersive communication experience combining virtual reality software and hardware, computer graphics and images, immersive communication, audio and video multimedia, and other technologies to ensure that users are able to experience being "immersed" [7]. Long-distance travel usually takes much time and vigor [8]. Many activities are dependent on people attending in person, such as conferences, large-scale competitions, lectures, etc. Today, there are many complete video conference applications or live broadcast applications (e.g., Skype and FaceTime) through which users can participate in remote activities via computers or mobile phones. These existing technologies or solutions make it possible for remote access to information from the perspective of results, but they are far from ideal from the perspective of the immersive experience effect due to a large gap with a real scenario.

Many online conferencing solutions already existed prior to the COVID-19 pandemic, such as multiperson video chat software, but they were far less popular than today. At that time, people were still willing to spend hours traveling to attend meetings that were sometimes less than an hour. The public is also willing to spend days buying tickets to performances or large-scale sports competitions. The reason is that a screen cannot provide the real feeling of immersion. Traditional online video conferences are widespread, and their enabling mechanisms are mostly based on traditional audio and video streams, which allow the main functions of the conference to be realized: video communication, voice communication, and presentation (such as slides). However, the gap between a video conference and an in-person conference is always the fact that is most criticized. This is because communication between people is not merely the simple exchange of information, and there are many other deeper behavioral exchanges that are important information in communication, such as body language, eye contact, gaze perception, etc. [9]. To achieve this sense of reality and immersion, virtual reality technology and equipment are needed. In this paper, we investigate the virtual reality conference room solutions, introduce the research state of the art, and propose a simple virtual conference room prototype that we have implemented.

Compared with traditional video, panoramic video can enhance immersion and bring an unprecedented visual experience. At present, the usage of panoramic video in various fields has been studied [10–12], and it is likely to be widely and deeply applied in the foreseeable future [13]. The panoramic video stream brings new video presentation forms to the on-demand and live broadcast fields. Scenes can be experienced more immersively, with a free choice of the viewing angle. At present, panoramic live broadcast technology has been applied to large-scale performances and event scenes. Thus, the audience can not only feel the atmosphere of the scene but also experience the perspective of the athletes in first person. At present, research on panoramic video streaming is focused on the video [14,15], audio [16,17], and quality evaluation [15]. The main focus is on the video, which involves acquiring the video, projection splicing, coding transmission, and equipment presentation. In this paper, we outline the current state of research in addition to presenting a simple panoramic live broadcast system designed and implemented by us.

### 1.1. Methodology and Contributions

In this article, a literature review investigation on the current research on VR conferencing and panoramic live broadcast is presented. By analyzing the related works, we determine the features and obstacles of VR conferencing. The procedure and key technologies of panoramic live broadcasting are also presented as well. With the use of available tools, the simple implementation of two immersive systems is given, aiming to provide a research and application reference for VR audio and video transmission, which can guide subsequent relevant research studies.

### 1.2. Article Structure

The paper is organized as follows:

- Section 2: overview on background of VR conferencing and panoramic live broadcast.
- Section 3: features and obstacles of virtual conference designing.
- Section 4: procedure and key technologies of panoramic live broadcasting.
- Section 5: essential role of 3D audio in the immersive experience.
- Section 6: methods of realizing the two typical immersive scenarios.
- Section 7: the main points of the article are summarized.

## 2. Related Work

Typical scene applications of virtual reality audio and video communication technology include VR conferencing and panoramic live broadcast. Panoramic live broadcast was developed as early as 2016, which is also considered the "first year of VR". Now, 5G, VR, and 8K have come into the public eye. With the rapid development of VR software

and hardware technology, panoramic live broadcast has become increasingly widely used in various industries. In recent years, due to the impact of COVID-19, researchers and the industry have begun to pay a greater attention to the development, design, promotion, and application of VR conferencing to improve the experience of remote collaboration. A VR conferencing room is one such typical application scenario. In addition, the international standards organizations Moving Picture Experts Group (MPEG) and 3rd Generation Partnership Project (3GPP), as well as the Audio and Video Coding Standards Workgroup of China (AVS), have been carrying out standardization research and development work related to immersive media, involving transmission protocols, audio and video coding and decoding, immersive communication systems, and other aspects, especially in the VR field, where the goal is to achieve a six-dof (degree of freedom) immersive experience. The research state of the art of VR conferencing and panoramic live broadcast in the academic research field is discussed next.

### 2.1. VR Conferencing

The purpose of video conference is to facilitate the possibility of instant communication between participants who are not physically together, through using network and multimedia technology, for participants to appear to be physically together. The traditional approach is to transmit the video and audio data [18,19] of each participant in their different spaces to all other participants through a multipoint control unit (MCU). At present, the technology for video conference has become increasingly flawless and has been widely used in the current COVID-19 pandemic. However, as video conference cannot provide a sense of presence, an immersive experience cannot be generated, opening the way for the introduction of virtual reality conference [20]. So far, the research and application of VR conference rooms is still in the start-up stage of development and exploration, with no widely accepted systematic theory.

In [21], a virtual reality conference room based on the traditional MCU architecture was proposed, in which participants could be placed in a virtual space for communication. A three-dimensional image in a virtual environment was rendered by processing the two-dimensional video information of the participants. This method could display the real image of people in a virtual space, as well as the body information in real time. However, the participants in the images still wore a head display, which may reduce the effectiveness of communication.A novel network video conference system based on MR display devices and AI segmentation technology was proposed in [22]. This was a robust, real-time video conference software application that made up for the simple interaction and lack of immersion and realism of traditional video conference, allowing users to interact in a conference in a new way. However, similar to [21] mentioned above, the participants in the images wore a head display. The presentation of real-time images of human subjects lacked realism if the observer's position was much different from the camera.

In [23], a more complex virtual reality conference system was proposed, which involved the generation of avatars and animation, especially facial expression animation. The authors provided three feasible means of multiperson cooperation: VR painting, slide display, and model import. There was also the general design of the network module. To finish, the experience of wearing the head display device was evaluated, and the conclusion was that the experience of wearing the head display device was better. A novel video transmission scheme of a virtual space conference was proposed in [24], which was based on perceptual control. In the system, the perception of each participant was calculated, with the construction of a perceptual space matrix based on this, wherein the matrix parameters were used to control the codec, effectively encoding the video object and thereby reducing the total bandwidth. The same approach was applied to audio objects.

In [25], the methods for evaluating the authenticity of immersive virtual reality, augmented reality (AR), and hybrid reality were summarized. The conclusion was drawn following the analysis of a large number of relevant articles. Most methods consisted of a combination of objective and subjective measures. The most commonly used assessment

tools were questionnaires, many of which were customized and unverified. The existence questionnaire was the most commonly used one , which was usually used to evaluate the authenticity and participation of one's existence and perception.

The research on VR conferencing is still in its early stages. Ref. [26] discussed the features and obstacles found within virtual conference solutions (both 2D and 3D) through a systematic literature review investigation. As a result, 67 key features and 74 obstacles users experience when interacting with virtual conferencing technologies were identified. The current VR conferencing solutions mainly focus on the visual aspect, and researchers focus on the representation of avatars in the virtual space. Under the limited equipment conditions, it is difficult to achieve a realistic avatar that makes people feel good. The exploration of sound and interaction is relatively small. Based on the above papers, the current research on VR conference can be divided into the following aspects: the presentation of characters, the interaction form in the virtual conference room, transmission optimization, and the evaluation of the conference experience effect.

### 2.2. Panoramic Live Broadcast

Panoramic video is also called 360-degree video, and it can facilitate more immersive feelings in people than traditional video. Audiences can observe from different directions at the same position in the video scene and feel all the information around the scene. Since the video picture contains all the information around the scene, in order to ensure that the resolution of pictures from all angles is not lower than that of traditional video, there must be more data for panoramic video, and the demand for bandwidth is therefore larger [27].

The acquisition and production of panorama requires multiple fisheye lenses for picture acquisition, and the pictures of each lens are spliced to form a frame picture. The panoramic video image acquired synchronously is then presented on a spherical surface, but its data form is not suitable for storage, transmission, compression, or other forms of processing by classical methods. Thus, projection is required [14,28]. The projection of panoramic video is the process of mapping the three-dimensional spherical information to a two-dimensional plane. Several projection methods used in the development of the Joint Video Exploration Team (JVET) coding standards are illustrated in [14]. As the main carrier of panoramic video, the projected image contains all the contents of the captured picture, and the distortion should be reduced as much as possible during splicing to improve the quality of picture playback. At the same time, [14] studied the impact of different projection methods on coding efficiency according to existing experiments.

The resolution of a panoramic video is very high. A larger network bandwidth is therefore needed while also using traditional video compression methods. In order to reduce the pressure of the network bandwidth, it is necessary to optimize the coding method for projected pictures. The panoramic live broadcast system can be divided into three types according to different encoding optimization methods [15]: full-view video stream, viewport-based video stream, and tile-based video stream. The full-view video stream transmits the whole frame, using the same encoding transmission mode as the traditional video stream. This approach requires a high processor performance and network bandwidth. Viewport-based streaming provides a high-quality transmission for the part viewed by the user based on the position of the user's current viewpoint, while the transmission quality of the rest is low. On the client side, the streaming endpoint device detects the user's head movement and receives only the specific required video frame area, dynamically selecting the video stream region of the viewport and adjusting the viewport quality. In this way, the bitrate of the video stream can be reduced. The server stores multiple adaptive sets related to a user's direction and performs matching and viewport position prediction according to the network state. Ref. [29] proposed two dynamic viewport selection approaches, which adapted the streamed regions based on content complexity variations and positional information to ensure viewport availability and smooth visual angles for VR users. Tile-based coding technology is used to divide each frame in the video stream into multiple blocks. According to the location of the viewport, the resolution between

different blocks will be different. For example, ref. [28] proposed a video streaming system that used the divide-and-conquer method to separate the video space into multiple blocks and encapsulate them during encoding. Ref. [30] proposed a sight-guidance scheme based on tile-based coding aiming at minimizing the weighted sum of the average traffic load and users' watching preference derivation. MPEG-DASH's (Dynamic Adaptive Streaming over HTTP) SRD (spatial relationship description) [31] is used to describe the relationship between blocks in the 360-degree space and is tiled in the field of view (FoV). Ref. [32] designed a feasible panoramic stereo video live-broadcast framework based on the current situation, which is similar to our system below.

At present, research on panoramic video is in a mature stage. Now and in the future, researchers will still explore better approaches to reduce transmission bandwidth and improve image quality. However, the research on the 3D sound effect of panoramic video needs to be expanded. Meanwhile, more research on applications on panoramic video is required as a supplement.

## 3. VR Conferencing

### 3.1. What a Virtual Reality Conferencing Application Needs to Succeed

As mentioned above, a virtual reality conference room is proposed with the aim of obtaining a feeling of immersion for users that is lacking in traditional video conference rooms. The design of the system focuses on this.

- Virtual space: Immersion means the users' experience of the scene. A virtual scene is the main part of the visual information and is an important source of immersion. The layout of the scene should conform to the appearance of the conference room, with tables and chairs for virtual avatars to move and interact. So far, there are many modeling tools as well as models made by others that can be obtained, which can be easily imported into the game engine for use.
- Avatar: Facial presentation is the most important and complicated factor in this section. One way to achieve this is to update the avatar's facial animation in real time by capturing the participants' facial information in real time, such as the eye rotation, lip movement, etc. When a user wears a helmet-mounted display, it becomes difficult to capture the face data, and the action of the face needs to be controlled by the content of the participants' speech [23]. As multiple participants join the conference room, each participant can see the virtual avatar of all other participants. When a user's position status or body movements change, the new status should be synchronized with other users, for which a network module that synchronizes the status of all players is required.
- Audio sense of space: Vision and hearing are presently critical aspects of bringing immersion to virtual reality devices. In order to obtain the same real experience as in the real conference room, auditory perception is as important as vision. To realize realistic auditory perception, spatial audio technology should be applied to virtual reality conferences. When in the same virtual conference room, each participant should be able to correctly sense the orientation of the other speakers as well as the reverberation of sound after multiple reflections from the room walls. While novel interactions are designed in the conference scenario, users can perceive the sound effect generated by the interaction when some interactions occur.
- Interaction: Communication in the meeting room should be supplemented by content presentation, such as slides, pictures, or videos. A virtual screen can be set up as the display area in the scene. In addition to basic interactions, there are many possibilities for interaction. In [23], VR painting and model import were mentioned. There are also new attempts to interact among participants.

### 3.2. Current Challenges

Nowadays, the technology of building virtual reality conference rooms is still incomplete and faces many challenges [26].

The challenge of network delay still exists. In contrast to a traditional video conference, there is no need to transmit the video stream of each participant, so the total bandwidth is reduced. However, to obtain the above spatial audio effect, it is necessary to transmit the audio stream of other participants to each participant while the system simultaneously updates the position, posture, and other information of all other participants for all users. The bandwidth demand is therefore still very large. Thus, a feasible scheme [24] is needed to reduce the bandwidth of audio transmission.

Another challenge is the lack of avatar authenticity. The user's impression of an avatar's posture and animation is still quite different from the real situation. The methods for avatar generation include avatars based on the user's real image and pure virtual avatars. An avatar based on the real image has a more realistic visual effect, but expensive equipment is required to achieve the ideal effect. Although completely virtual avatars cannot show the real appearance of participants, they tend to be more integrated with the virtual environment, and the body animation is more natural [33].

There is no available solution for interaction design. Traditional video conference is the mainstream form at present and will likely remain so for a long time. One reason is that its interaction design is mature, and we can experience similar interaction modes even using different video conferencing software, thus it could save us time in getting familiar with new software. In such circumstances, virtual reality conferencing has a long way to go [34]. Moreover, unlike familiar operations on computers, participants usually need more time to get familiar with their interaction in a virtual reality environment.

How to attract more people to virtual reality conference rooms is also a critical challenge. In [8], it was mentioned that the virtual conference itself reduces the opportunities for interaction between people. Another viewpoint is that important meetings should be face to face, and people can better prepare for the meeting by taking advantage of the long-distance travel time [35].

## 4. Panoramic Live Broadcast

### 4.1. Steps to Implement Panoramic Live Broadcast

In October 2015, the standardization of the panoramic video packaging format was launched by MPEG, called omnidirectional media format (OMAF) [36], which was jointly developed by experts from high-tech enterprises, research institutions, and universities around the world.

It is stipulated in the OMAF standard that a panoramic video spliced on the server side can be streamed using a DASH or MPEG media transport protocol after content preprocessing, encoding, and encapsulation. On the one hand, an OMAF player can receive, parse, and present relevant media content; on the other hand, it needs to track a user's head movements, eye movements, and other interactive operations to feed back window information in real time. Figure 1 shows the panorama video processing flow specified in the OMAF standard.

Shooting and splicing: A panoramic camera is required to obtain panoramic videos, whose information is sourced from a composite of multiple cameras. In cases where a higher video resolution is required, more cameras of higher resolution are needed. Stitching technology is used to place the pictures of all cameras around the observer to generate the panoramic picture. If there are six ideal identical cameras, they should be placed on a bracket with a strict position and rotation to take pictures in four horizontal directions and in the up and down directions; thus, pictures that are taken should be spliced together to form a square. In that case, the results can be used directly without optimization. However, it can be difficult and even impossible to accurately meet the above standards. Therefore, some researchers use methods [37] to reserve image redundancy and correctly identify and process the areas where the images of two cameras overlap and then splice them.

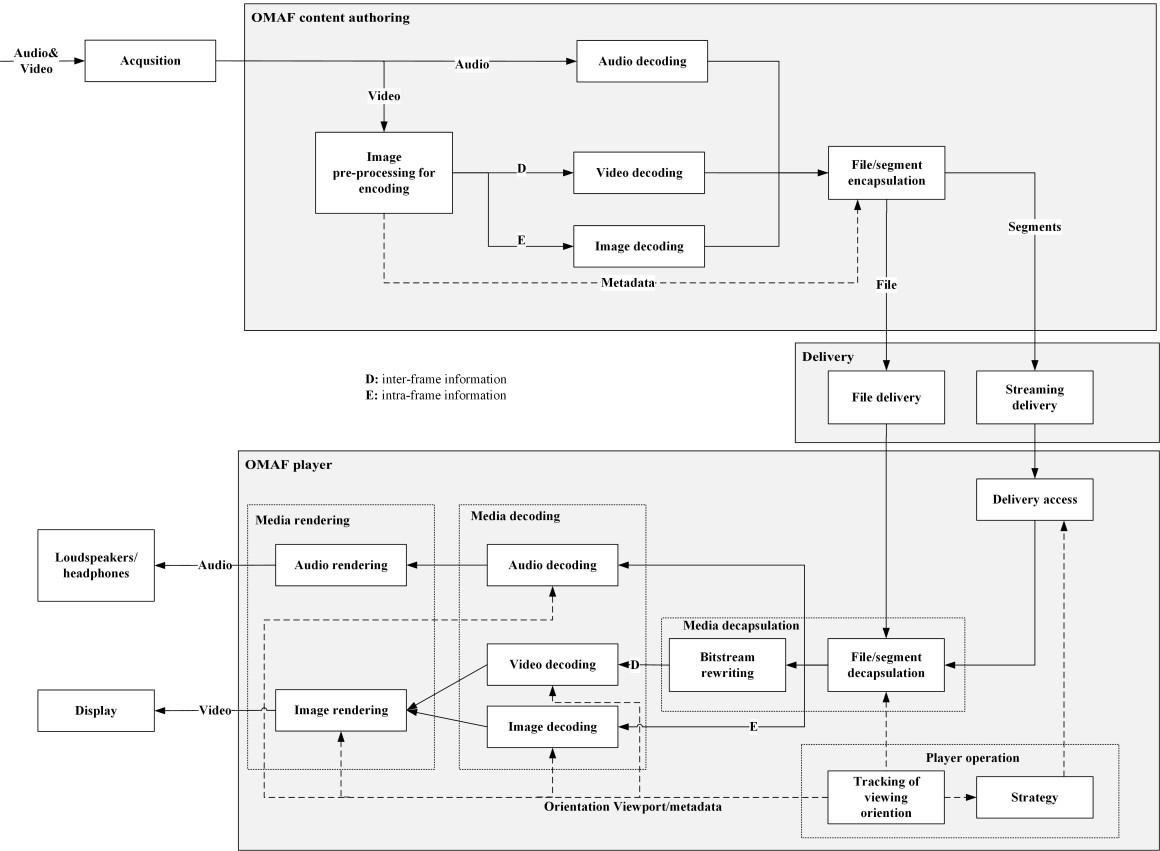

**Figure 1.** The OMAF architecture for panorama video processing flow. Three main steps are involved: content authoring, delivery, and playing. Content authoring is completed by a server, the delivery is over the Internet, and the content obtained is generally displayed on a personal computer [36].

Projection and back projection: Each frame of a panoramic video is presented in the form of a sphere. However, the original video coding standards are applied to the transmission of panoramic video. That means there should be preprocessing before transmission. The process of mapping the spliced picture to a rectangular picture is called projection. At present, the OMAF standard only supports longitude and latitude maps, equirectangular projection (ERP), and cube mapping (CMP) [36]. ERP is similar to the generation of the world map. In ERP, people look from the center of the sphere outward to the surface of the sphere inward, while for the world map, they look at the sphere from the outside to the inside. In CMP, the complete sphere is divided into six regions, which are projected onto each of the six faces of the cube. The bottom, back, and top faces need to be arranged together with the other three faces to form a rectangular frame through a specific rotation operation. In order to improve the coding efficiency, the operation principle of the three rotating surfaces is to maintain the consistency of the media content at the junction of the arrangement time surface and the surface [36].

Encoding and decoding: Traditional video encoding and decoding schemes may meet the codec requirements of panoramic video streams. However, because panoramic video itself is different from traditional video, many problems arise when traditional schemes are applied to panoramic video, such as geometric distortion, discontinuous pictures, etc. As a result, various optimization schemes for panoramic video codecs have been put into practice [38,39].

Streaming: It was described in the second part of the related work that there were three mainstream panoramic live broadcast schemes: full-view video streaming, viewport-based video streaming, and tile-based video streaming. Determining how to make a better choice between improving video quality and reducing transmission bandwidth is still the main direction of current and future research.

*4.2. Current Challenges for Panoramic Live Broadcast*

Panoramic video provides an immersive volume of video that is not available in traditional 2D. A panoramic video is spherical in nature, which brings about many challenges regarding its acquisition, storage, coding, transmission, and display.

There is a variety of distortions from capture to display. To solve the distortion problem in 360-degree video streams, efficient splicing, projection, and encoding methods with better exploration results and a lower bandwidth should be explored. With the popularity of virtual reality technology and the formulation of the next-generation video coding standard, 360-degree video has been receiving increasing attention from academia and various fields in industry. Innovating video projection methods and efficient compression coding to meet bandwidth and quality requirements represent the current direction of mainstream research and exploration [14].

Sphere-to-plane projection is a common procedure for panoramic video before encoding. There are, so far, many projection formats. Considering that different plane projection formats may be adapted to different applications, we often need to convert from one projection format to another, for which the interpolation algorithm is thus critical [40].

It is necessary to focus on designing quality evaluation methods and indicators for panoramic video. This is a complex challenge, for traditional video QoE (quality of experience) models are not suitable for 360-degree content. Although most studies have carried out various subjective and panoramic video objective evaluations, most evaluation methods still follow the traditional video evaluation standards. Due to the lack of unified and standardized factors affecting 360-degree video, the standard evaluation method has not been finalized. This represents a complex and challenging problem [15].

## 5. The Importance of VR Audio

Also known as virtual audio, spatial audio, and immersive audio [41], 3D audio includes two forms of playback based on binaural audio and speaker playback. The former is widely used in VR devices, usually called binaural audio. The basic principle of binaural audio is to simulate the sound field generated by a sound source at a certain point in space in two ears such that the listener can have a sense of where the source was emitted from. This technology is also known as binaural acoustic technology [42,43]. Because of the inherent characteristics of the human auditory system, there are often certain differences between the sound that people subjectively feel and natural sound. The effect of the human auricle and other structures on sound waves can be seen as a filter, called head-related transfer function (HRTF). HRTF simulates the human ear's perception of sound direction and distance in space, which plays an important role in the generation of binaural virtual audio.

With the development of virtual reality devices, more and more immersive applications are emerging. Although most research is focused on how to improve the resolution and frame rate of frames, hearing plays as equally an important role as vision in providing immersion [44,45]. Audio is one of the important outputs of electronic games [44]. In first-person shooting games, players need to identify the position of the enemy according to the direction of the sound for an effective reaction. In addition, game audio can play an important technical role in providing basic user feedback by providing player behavior confirmation or warning of in-game activities. Research in [46] showed that in a shared space, with the inclusion of spatial audio and video, users can identify speakers better, retain more information, and have an increased comprehension from video conference meetings.

As is indicated in [47], sound may have a greater influence on immersion than immersion has on images. Sound is one of the most important sources of human cognition. VR/AR content matching of audio effects has a great impact on enhancing users' memory of content. Dale, an American audiovisual educator, put forward the theory of a "tower of experience" in his book *Audio-Visual Methods in Teaching*. This theory considers how human experience is derived. Figure 2 shows the top three layers in Dale's tower of experience (the main theoretical basis of audiovisual education). The third row from the top represents

audio/recording/photos, and the layer above is visual signals and language symbols, indicating that the combination of listening and audition is more conducive to memory and understanding of knowledge.

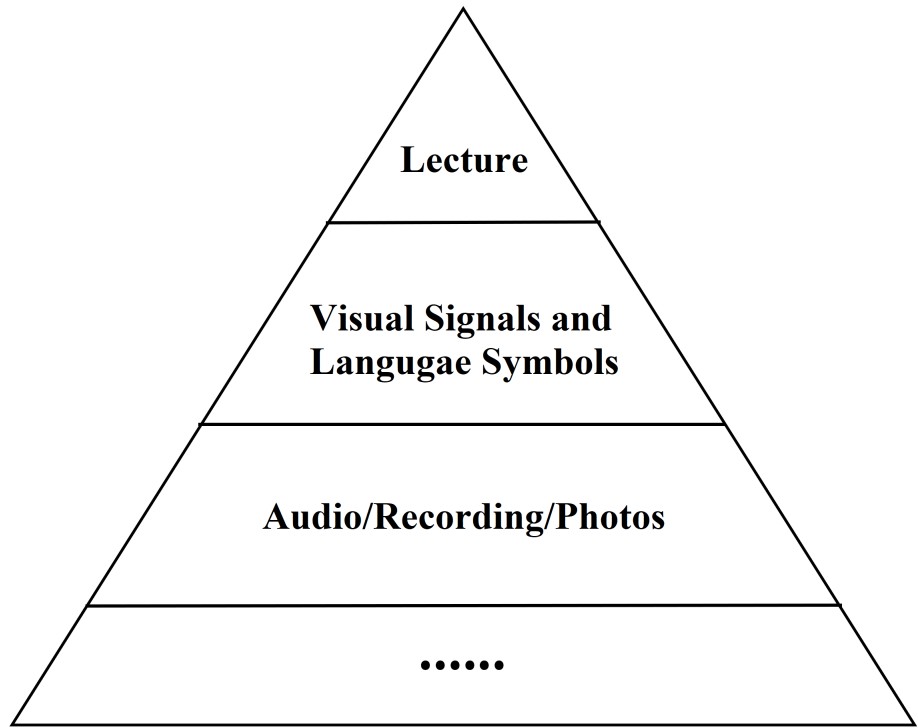

**Figure 2.** The top three layers in Dale's tower of experience [48].

The demand for VR will increase along with the improvement of hardware, and the demand for immersive audio is also growing. It is pointed out in [49] that we need better authoring tools to support the creation of high-quality immersive audio works, regardless of whether the creators understand the underlying principles of audio, just as many video creators do not understand the underlying codec of a video.

## 6. Methodology of Sample Implementation

Two simple implementations for the above two scenarios are presented in this section, while the implementation of each module or step is introduced. The results are given at the end of each part.

### 6.1. Implementation of VR Conferencing

Now, we introduce a simple virtual reality conference room, which includes a scene with audio and video experience created by a Unity engine and simple network communication.

6.1.1. Architecture

The overall architecture of the system is shown in Figure 3.

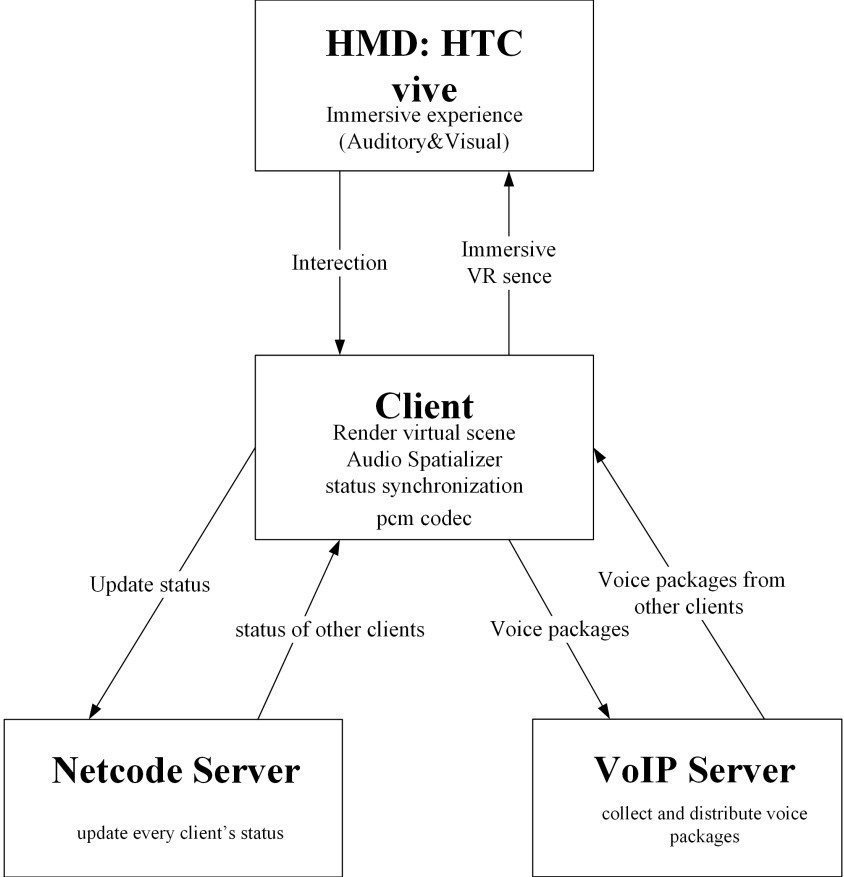

**Figure 3.** Architecture of the virtual reality conference room. The two servers are for Netcode and VoIP, respectively. The spatial audio effect is realized on the client computer. The immersive feeling is generated by a HMD.

The whole system is comprised of mainly three parts: server, client, and a HMD.

- Server: We use two servers, a Netcode server and a VoIP server. Netcode is a concept in game development. The Netcode server in our system is to synchronize each user's status, allow the user to see the precise and fluid representation of the room state, and influence the scene state shared in common. The VoIP server controls the transmission of audio packages on network. It receives voice packages from each client and distributes them to each other clients.
- Client: The client, usually a high-performance computer, is responsible for collecting the microphone input from the HMD, obtaining the pulse code modulation (PCM), compressing the PCM packets, and sending it to the VoIP server. Meanwhile, it receives the PCM packets from other clients sent by the server, decompresses them, and processes them with the spatial audio algorithm [50] to play the audio with room reverberation and orientation. The client synchronizes the position to the Netcode server and updates the position of other clients obtained from the server.
- HMD: The virtual scene is rendered in real time and could be experienced in an HMD, which meanwhile transmits a user's physical state and interaction data to the client. In our system, we used HTC Vive as the HMD shown in Figure 4.

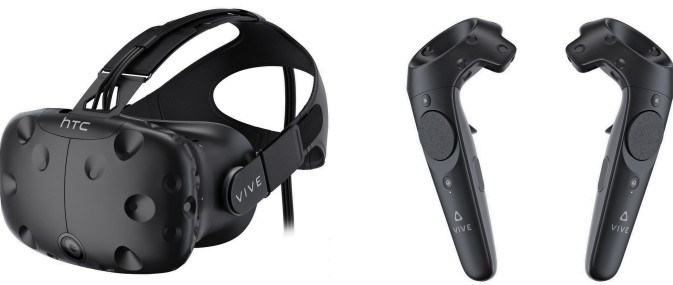

**Figure 4.** HTC Vive.

6.1.2. Scenes, Avatars, and Interactions

To simplify the workload, the modeling of the scene [51] and avatar [52] were from free resources on the Internet. Figure 5 shows how our VR conferencing room looks. Figure 6 is the avatar of the user. After a participant enters the virtual conference room, they are assigned to a random seat. At this time, they can change the direction of view and observe the surrounding environment by moving the mouse or rotating the HMD (head-mounted display). When another participant also enters the conference room, the two participants can see each other in the scene and talk to each other by voice. The view of a user is shown in Figure 7. The sound effect produces a real sense of space and orientation. When there are multiple participants in the conference room, everyone can feel the orientation of other participants.

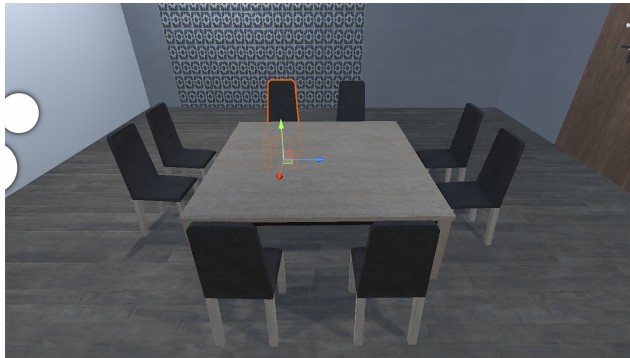

**Figure 5.** Virtual conference scene.

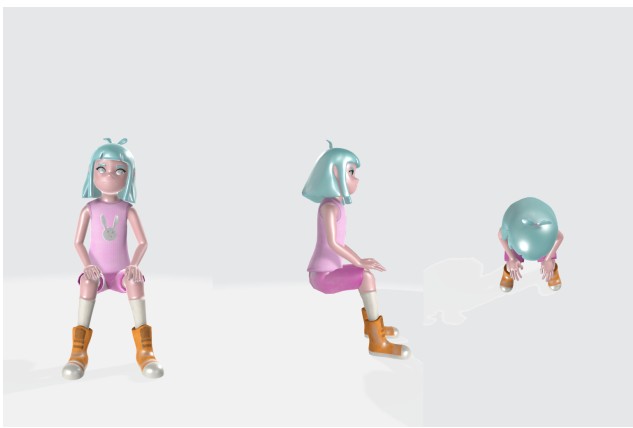

**Figure 6.** Avatar.

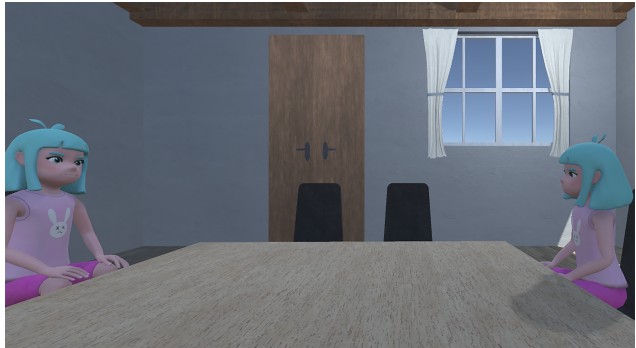

**Figure 7.** View of a user.

### 6.1.3. NetCode for Status Synchronization

In online games, in order to allow multiple players to play games together on different computers, a mechanism is needed to ensure that all computers are synchronized such that players can accurately see the performance of each player smoothly; players' input should influence the game state. We use the NetCode [53] network architecture of Unity to synchronize the status information of roles in multiple terminals, mainly including location and orientation information. The network code has always been one of the most difficult parts in game development. The Unity NetCode software package provides a dedicated server model with client prediction, which can be used to simplify the development of multiplayer games. It is currently in the experimental phase. Listing 1 is some pseudocode for Netcode.

**Listing 1.** pseudocode of Netcode in unity

```
1   Vector3 Position = new Vector3;
2
3   Move()
4   {
5       if (this.IsServer)
6       {
7           newPosition = GetPosition();
8           Position = randomPosition;
9       }
10      else
11      {
12          SubmitPosition();
13      }
14  }
15  [ServerRpc]
16  SubmitPosition()
17  {
18      Position.Value = GetPosition();
19  }
20
21  void Update()
22  {
23      transform.position = Position.Value;
24  }
```

### 6.1.4. Simple VoIP

The traditional multiperson VoIP voice conference refers to a server that obtains the voice code stream of each client, synthesizes it, and sends it to each client. In order to

realize the sense of direction of each participant's voice, the voice data stream of each participant should not be merged on the server but, instead, separately distributed to each other. Without lowering the voice quality, this method requires more bandwidth than traditional voice conferencing. Moreover, as the number of participants increases, the pressure on network bandwidth grows rapidly. Both the server and the client have to bear the pressure of the network bandwidth, and the demand on server grows faster. The network pressure can be relieved by limiting the number of participants. As the local area network (LAN) environment is in good condition, we simplified the VoIP process into the following sequence: collect audio signals, encode and package them, serialize them, send them to the server, transmit them to other clients, and the client receives a packet, deserializes it, unpack and decode it, and play the audio. We used the User Datagram Protocol (UDP) for network transmission and the Speex open source library [54] for the encoding and decoding.

Listing 2 is some pseudocode for the network transmission.

**Listing 2.** pseudocode of VoIP

```
1  // voip server
2  sendData(data)
3  {
4      //send audio package to every client
5      for (int i = 0; i < client_count; i++){
6          if (hasClient[i]){
7              mySocket.SendTo(..., Remotes[i]);
8          }
9      }
10 }
11
12 //voip client
13 startSocket()                          //start udp socket
14 {
15     mySocket = new Socket(UDP);
16     IPEndPoint sender = new IPEndPoint();
17     Remote = (EndPoint)sender;
18     int recv = mySocket.ReceiveFrom(data, Remote);
19     while (true){
20         //receive audio package from server
21         mySocket.Receive(data, Remote);
22         process(data);               //process audio data
23     }
24 }
```

### 6.1.5. Result

In the VR conference room solution given above, the basic functions of the conference were achieved. Users could switch perspectives horizontally, make speeches, and communicate with other participants. In terms of listening feeling, users heard the voices of different participants from different directions and experienced the effect of room reverberation. The NetCode latency and VoIP latency in the LAN were acceptable. Character animation and scenes could be optimized by referring to 3D games, and more applicable interactive functions could be further explored.

### 6.2. Implementation of Panoramic Video Broadcast

#### 6.2.1. Architecture

Figure 8 shows the architecture of our panoramic video broadcast system.The following are the main procedures for making panoramic live broadcasts: audio and video

capture (Figures 9 and 10), projection (Figure 11), compression, encapsulation, and data stream transmission. The receiver obtains the code stream and then decodes it. After backprojection transformation, the panorama video can be displayed. Next, is the process of implementing a simple panoramic live broadcast.

Video capture:

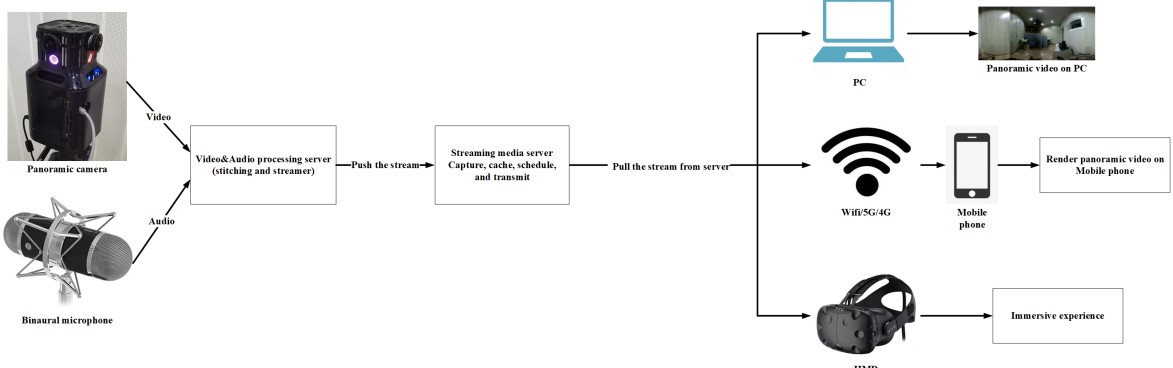

**Figure 8.** Architecture of panoramic video broadcast system. This is a simplified implementation of Figure 1. The cameras and microphone are used to capture the original input. Through the simple processing of the server, the content can be distributed to various terminal devices.

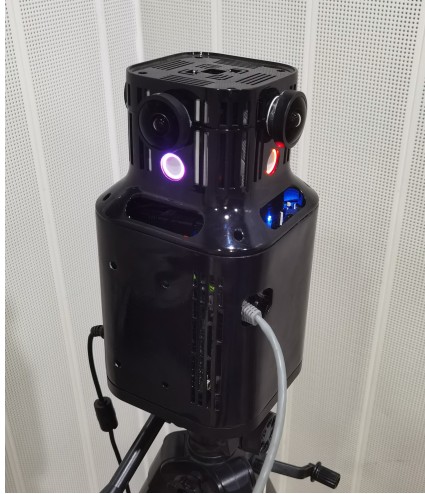

**Figure 9.** Four fisheye camera, integrated with projection transformation, compression coding, and network transmission modules.

Audio capture:

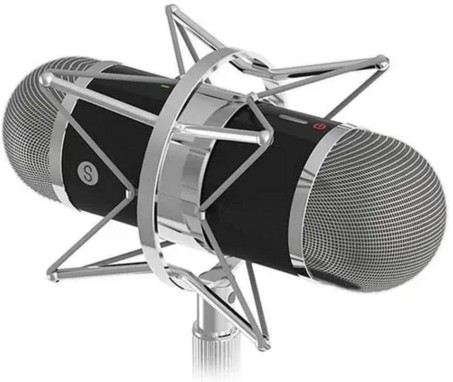

**Figure 10.** A binaural microphone that records a directional dual-channel audio stream. The sound direction is determined by its orientation.

Projection:

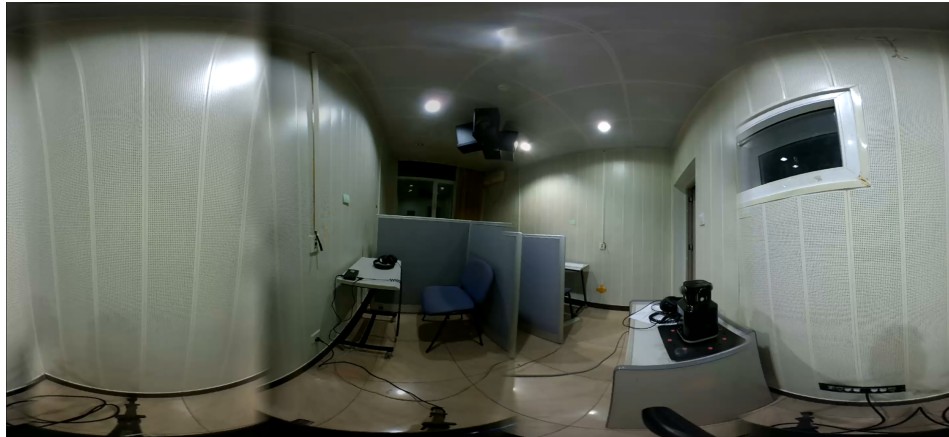

**Figure 11.** Four pictures of one frame spliced into one plane for coding and compression.

6.2.2. Network Transmission

For the network transmission protocol part, RTMP (Real-Time Message Transmission Protocol) was adopted. The 2D images after projection splicing and audio coding were encapsulated and then pushed to the network. On the client side, the corresponding RTMP stream was pulled. The content in Figure 11 is also the real-time frame of RTMP stream obtained by the client through testing. There was a sense of direction of the sound, but it was still difficult to achieve a three-dof sound direction change through the perspective transformation.

6.2.3. End-to-End System Delay Analysis

Based on the panoramic video live broadcasting system, we conducted an end-to-end delay analysis, and the system set the duration of each frame at 30 ms.

(1) The camera captured and cached one frame, resulting in a 30 ms delay;

(2) The splicing box could collect up to three cached frames and generated a maximum delay of 90 ms;

(3) The maximum splicing buffer was three frames, resulting in a maximum delay of 90 ms;

(4) The maximum encoding buffer was two frames, resulting in a maximum delay of 60 ms;

Therefore, the maximum delay was 270 ms at the VR panoramic video content production end.

(5) The delay was determined by the bandwidth and the bit rate of the transmitted video. We use a wired local area network in the laboratory according to the previous implementation. In the case of the network test, when the overall minimum delay was 500 ms–1 s, the minimum delay of this part was within 500 ms, although wireless transmission would be higher;

(6) + (7) + (8) The delay at the playback end was less than 20 ms for mainstream display devices. Network transmission delay was still the main factor causing end-to-end delay. After the above analysis, the following system (Figure 12) delay analysis composition diagram was obtained.

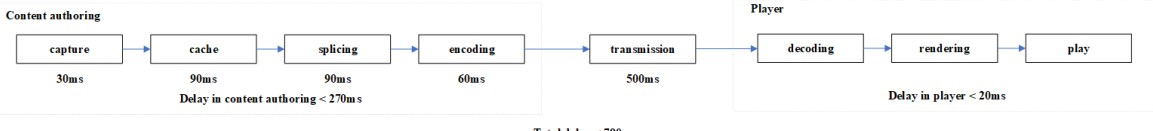

**Figure 12.** System delay analysis. The delay mainly comes from the content authoring and transmission, and the delay proportion of the player is very small.

## 7. Conclusions and Future Work

As virtual reality technology and equipment develop, the demand for immersive experience will continue to grow. We presented two main patterns of immersive experience. One was the pure virtual environment represented by games, in which users can interact. Most of these scenes are created by game engines such as Unity or Unreal. The other was the panoramic video obtained by the processing of real scenes captured by cameras. In this article, we discussed these two kinds of immersive audio and video scenes and provided sample implementation approaches for these two systems.

We discussed the design and implementation of virtual reality conference environments, outlined the current research status in this field, and analyzed the many challenges. Then, we designed and implemented a simple virtual reality conference room, for which the virtual scene design, audio transmission, and realization of the spatial orientation were described. In the future, research will increasingly focus on how to generate more realistic avatars as well as facial expression optimization. There will be more exploration of the interaction patterns in conference rooms. With the development of virtual reality hardware equipment, new forms of interactions in virtual reality conference rooms may arise. In future work, discussions about the advantages and disadvantages of VR conferencing [8,35,55] will continue.

In this article, we built a panoramic live broadcast system. By designing the system architecture and completing each module, we implemented a panoramic live broadcast system. The delay analysis was presented. In the future, we will do more research based on this system. Today, panoramic video live broadcast is widely used, especially in some live events. How to use a lower network bandwidth to achieve a better playback effect will be the focus of future studies. Since panoramic video is presented as a sphere, the client playing the video needs to project the picture onto the sphere, which imposes certain requirements regarding the rendering speed of the computer. Future research should consider how to improve the rendering speed and reduce the consumption of performance on the premise of ensuring that the picture is not distorted. In addition to vision, the source of immersion of panoramic video should be the sense of the audio space. It is worth paying attention to facilitating users' experience in sensing the audio space in the panoramic video. Specifically, the content that is suitable for a traditional video presentation may not be suitable for panoramic video and vice versa. We should explore more appropriate content for panoramic video, which may also improve demand for panoramic video.

**Author Contributions:** Conceptualization, H.Z. and J.W.; methodology, H.Z. and J.W.; software, H.Z.; validation, H.Z., J.W. and Z.L. ; formal analysis, H.Z.; investigation, H.Z. and J.W.; resources, J.L.; data curation, H.Z. J.W. and Z.L.; writing—original draft preparation, H.Z.; writing—review and editing, H.Z.; visualization, J.L.; supervision, J.L.; project administration, H.Z.; funding acquisition, J.L. All authors have read and agreed to the published version of the manuscript.

**Funding:** National Natural Science Foundation of China (grant no. 62071039) and Beijing Natural Science Foundation (grant no. L223033).

**Data Availability Statement:** Not applicable.

**Acknowledgments:** The authors wish to thank all participants who volunteered for this study. Informed consent was obtained from all subjects involved in the study.

**Conflicts of Interest:** The authors declare no conflict of interest.

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
