# Peer review of "Design and Implementation of Two Immersive Audio and Video Communication Systems Based on Virtual Reality"

_electronics, doi:10.3390/electronics12051134_

Round 1

Reviewer 1 Report

This paper proposed “Design and Implementation of Immersive Audio and Video Communication System Based on Virtual Reality”. The approach discussed in this manuscript is interesting. I recommend following corrections.

1-     Related work needs more investigation of some latest and relevant work

2-     Briefly described your proposed methodology

3-     Clearly defined your manuscript motivation and practical application

4-     All figure revised and improved visual quality of images

5-     Add Pseudocode Examples of your proposed and based method

6-     The experience section provides additional information, justifies your work, and compares it to other states or arts methods. 

7-     Most of the references in the Bibliography sections are quite old. Recent references may be used.

8-     Needed extensive proofreading.

Reviewer 2 Report

Sections 1 and 2 introduce the paper and the related works. Those two sections are well-written, provide enough context to ease readers’ comprehension. The carried out literature review is adequate and supplemented with recent literature. Section 3 describes in detail what virtual reality (VR) conferencing is, what it can be used for, as well as its current challenges. Section 4 presents the steps needed to implement panoramic live broadcast. This is aided with the addition of a figure that displays the OMAF architecture. The challenges facing panoramic live broadcast are also outlined. Section 5 discusses the importance of VR audio, or 3D audio. Section 6 is the heart of this manuscript, where sample scenarios and all the technical details of the proposed VR conferencing system are provided. This is aided with the provision of related images and figures. In this section, 2 different forms of VR conferencing are showcased. The conclusions are finally provided in Section 7. These summarize the main ideas of the manuscript, highlighting its most important aspects, as well as emphasizing the significance and timeliness of the topic of VR conferencing.

Overall, this is a well-written manuscript, with rather few language mistakes. Although it would have been better to include more information about the technical aspects of its implementation, including the architecture, network protocols, infrastructure needs and so on.

Reviewer 3 Report

See attached comments

Reviewer 4 Report

For a better manuscript, please clearly present the results and conclusions in the abstract.

Round 2

Reviewer 1 Report

The author didn’t carefully read the comments and give the answer, in the first revision still needs improvement and mention where you did fix the problem in the final manuscript, you should highlight. 

1-     Add more detail in the captions of figures 1, 3,8, and 12 so that they are self-explanatory to the reader. 

2-     Even if it's not compulsory, I suggest you upload your source code with the revised manuscript as an additional file.

3-     There are ambiguities and unclear meanings throughout the whole paper. Be more precise with your language.

Reviewer 3 Report

See comments for minor changes

Round 3

Reviewer 1 Report

Before submitting the manuscript, kindly show comments to the corresponding author/ supervisor and respond according to the reviewer's comments. Add more detail in the captions of figures 1, 3, 8, and 12 so that they are self-explanatory to the reader.  
